

# Long-term legacy of phytoremediation on plant succession and soil microbial communities in petroleum-contaminated sub-Arctic soils

Mary-Cathrine Leewis[1,2,3], Christopher Kasanke[2,3,4], Ondrej Uhlik[5], Mary Beth Leigh[2,3]

5   [1] Agriculture and Agri-Food Canada, Quebec City, G1V 2J3, Canada
    [2] Institute of Arctic Biology, University of Alaska Fairbanks, Fairbanks, 99775, USA
    [3] Department of Biology and Wildlife, University of Alaska Fairbanks, Fairbanks, 99775, USA
    [4] Walla Walla Community College, Walla Walla, 99362, USA
    [5] Department of Biochemistry and Microbiology, University of Chemistry and Technology Prague, Prague, 160 00, Czechia
10  *Correspondence to:* Mary-Cathrine Leewis (mary-cathrine.leewis@agr.gc.ca)





**Abstract.** Phytoremediation can be a cost-effective method of restoring contaminated soils using plants and associated microorganisms. Most studies follow the impacts of phytoremediation solely across the treatment period and have not explored long-term ecological effects. In 1995, a phytoremediation study was initiated near Fairbanks, Alaska, to determine how the introduction of annual grasses and/or fertilizer would influence degradation of petroleum hydrocarbons (PHCs). After one year, grass and/or fertilizer treated soils showed greater decreases in PHC concentrations compared to untreated plots. The site was then left for 15 years with no active site management. In 2011, we re-examined the site to explore the legacy of phytoremediation on contaminant disappearance, as well as plant and soil microbial ecology. We found that the recruited vegetation, along with current bulk soil microbial community structure and function were all heavily influenced by initial phytoremediation treatment. The number of diesel-degrading microorganisms (DDM) was positively correlated with increasing amounts of vegetation on the site, and inversely correlated with PHC concentrations. Even 15 years later, the initial use of fertilizer had significant effects on microbial biomass and microbial community structure activities. We conclude that phytoremediation treatment has long-term, legacy effects on the plant community, which, in turn, impacts microbial community structure, function, and continued TPH disappearance. It is therefore important to consider phytoremediation strategies that not only influence site remediation rates in the short-term, but that also prime the site for restoration of vegetation across the long-term.

**Graphical Abstract.** Photos of study site in 1996 (left, photo credit: C.M. Reynolds) and in 2011 (right, photo credit: authors) showing changes in plant communities in the different soil types and phytoremediation treatment plots.

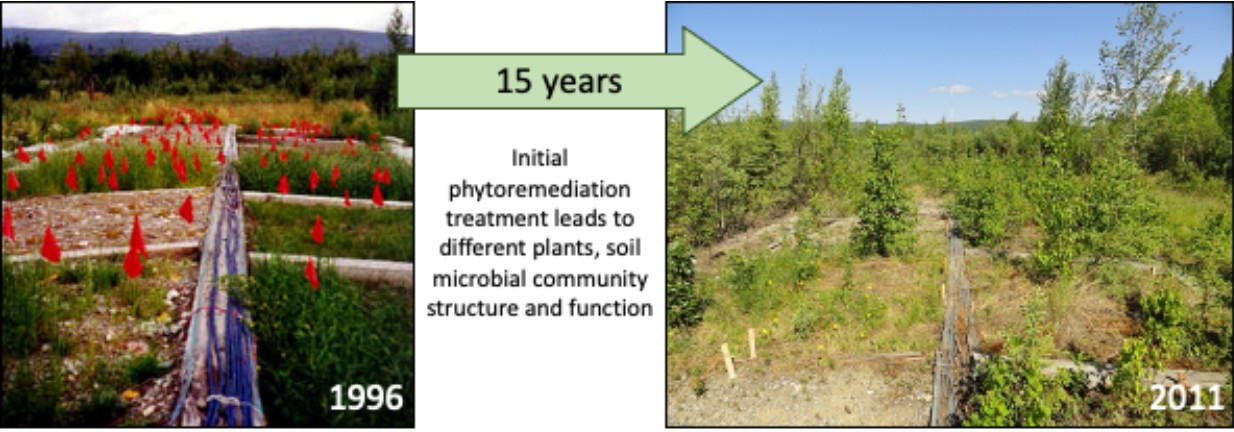

**Copyright Statement**



40

## 1 Introduction

There is a worldwide legacy of contamination from operational and abandoned mining and industrial sites, waste disposal sites, abandoned and active military installations, and other anthropogenic activities (Stow *et al.*, 2005; Pier *et al.*, 2002). In particular, PHCs are widespread in the environment as a consequence of their broad use in society, posing serious human health hazards. In the state of Alaska, there are nearly 2,000 documented sites with contaminated soil, many of which contain petroleum (Alaska Department of Environmental Conservation Spill Prevention and Response, 2015). The harsh climate and remote location of many communities at high latitudes pose unique challenges for contaminant remediation. Lack of road access can make traditional physical remediation methods difficult, cost-prohibitive, or even impossible (Kuiper et al., 2004). Bioremediation offers a potentially affordable and feasible solution for contaminant clean-up in remote communities which are especially dependent on petroleum products for heat and transportation. Soil amendments, such as fertilizer or plants, can be used to stimulate the degradation of contaminants by indigenous soil microorganisms and is often more cost effective in remote areas (Siciliano et al., 2003; Aken et al., 2010; Wenzel, 2009; Leewis et al., 2016). The biodegradation of organic pollutants in plant-associated soils is hypothesized to be driven by the release of secondary plant metabolites (SPMEs), which can be similar in chemical structure to the contaminant (Leigh et al., 2002; Singer et al., 2003; Musilova et al., 2016; Zubrova et al., 2021).

The microbial and vegetative processes associated with biologically-based remediation strategies, i.e. phytoremediation and rhizoremediation, have been a focus of study for decades (Gerhardt et al., 2009; Glick, 2010; Palmroth et al., 2002; Siciliano and Germida, 1998; Macek et al., 2009). In many cases, remediation treatment results in decreases in soil contaminant concentrations and changes in the soil microbial community (Liu et al., 2015; Yergeau et al., 2015; Bell et al., 2013; Palmroth et al., 2005; Leewis et al., 2016; Lopez-Echartea et al., 2020). Many of these studies, however, focus on the short-term effects of treatment on microbial and vegetative communities and contaminant transformation. Monitoring the long-term effects of contamination and subsequent treatment on soils, beyond the first few years of active site management or experimentation, is important to assessing the long-term impacts on ecological processes such as plant succession and microbial community function (Susarla et al., 2002; Mukherjee et al., 2014). Few studies have investigated the long-term

65   effects of remediation treatments on PHC-contaminated soils; most focused on the effects of land-farming on contaminant

concentrations (Loehr and Webster, 1996; Mills et al., 2006) without determining vegetation recruitment or effects of treatment

on microbial communities. Other studies have explored the long-term effects of treatment and soil chemistry on microbial

communities (Sutton et al., 2013; Liang et al., 2016), yet without determining the effects of treatment on vegetation

recruitment. While these studies provide insight to the disappearance and biodegradation of legacy PHC contamination, even

70   these comprehensive investigations stop short of fully characterizing the *in-situ* degradation and plant-associated changes in

microbial communities that would be necessary for an integrative understanding of the effects of different remediation

strategies on site ecology. More long-term studies of ecosystem recovery are essential to achieving an integrated understanding

of how plant communities, remediation treatment, contaminant concentration, and microbial populations interact as sites

change over time.

75        Here, we seek to further understand the long-term (over-15-year) relationships between original phytoremediation

treatment, colonizing vegetation, soil microbial communities, and the degradation of petroleum contaminants at an

experimental site in interior Alaska (Leewis et al., 2013; Reynolds et al., 1997a; Reynolds and Koenen, 1997). The original

phytoremediation study by Reynolds and colleagues was designed to investigate the effects of combinations of planting and

fertilizer treatments on the remediation of soils contaminated with either crude oil or diesel fuel (Reynolds et al., 1997a;

80   Reynolds and Koenen, 1997; Reynolds et al., 1999). The site was re-examined to assess the effects of the original treatment

on hydrocarbon loss, plant community composition, and the bacterial rhizosphere and endosphere communities after 15 years

without active site management (Leewis et al., 2013; Papik et al., 2023). In the current study, we build upon this previous work

and aim to i) evaluate the interactive effects of petroleum contamination and phytoremediation on site recovery over time; and

ii) identify the factors that drive vegetation community structure, soil microbial community structure and function, and total

85   petroleum hydrocarbon (TPH) loss.



## 2 Methods

### 2.1 Legacy Field Study Background and Site Description

In 1995, a phytoremediation field study was initiated by the Army Corps of Engineers Cold Regions Research and Engineering Laboratory (ACE CRREL) located in Fairbanks, Alaska (Reynolds et al., 1997c, 1999; Reynolds and Koenen, 1997). The study sought to compare the effects of different phytoremediation treatments on two soil and contamination types: A) crude-oil-contaminated soils collected from a gravel pad, or B) diesel-contaminated soils collected during the removal of an underground storage tank. The experiment was initiated at the ACE CRREL Farmers Loop Facility, where soils were placed in piles in adjacent lined areas. Each of the separate soil piles was then divided into individual plots (Fig. 1) which represented different phytoremediation strategies. The original phytoremediation treatment plots were; two plots not planted or fertilized (c1 & c2), one plot spread with fertilizer (f), one plot planted with annual ryegrass (*Lolium multiflorum*; p1), one plot planted with a mixture of annual ryegrass and Arctared fescue (*Festuca rubra*; 1:1 mixture, p2), one plot fertilized and planted with annual ryegrass (p1f), and one plot fertilized and planted with both ryegrass and fescue (p2f; Fig. 1C). Fertilizer was applied to soil surfaces by hand at approximately 620 g/m$^2$ of N, P, and K using a commercially available fertilizer (granular 20-20-10). Seeds were spread by hand at approximately 10.8 g/m$^2$.





**Figure 1. Overview of Site. Photos of study site in (A) 1996 (photo credit: C.M. Reynolds) and (B) in 2011 showing plant colonization of the different soil types and phytoremediation treatment plots. C) Overview of the original phytoremediation treatment plots (outlined in thick black lines). Half of the plots are soils from a gravel pad originally contaminated with crude oil (white background) and half are soils from near a leaking diesel fuel storage tank contaminated (grey background). The original treatments applied in 1995 were as follows: no treatment (c1, c2), planted with annual ryegrass (p1), a mix of annual ryegrass and Arctared fescue (p2), treated with fertilizer (f), and/or no added nutrients (no "f" indicated). For this follow-up study, original plots were each subdivided into six subsections (outlined in dashed grey lines) to allow for pseudo-replication.**

After 238 days of treatment, TPH concentrations in soils treated with plants or fertilizer (p, f, p2) and soils without any amendment (c1, c2) decreased relative to the initial (day 0) TPH measurements (Table 1). Soils treated with both plants



and fertilizer (pf, p2f) had significantly lower TPH concentrations than the untreated soils (c1, c2) after 238 days of treatment (Reynolds et al., 1997a; Reynolds and Koenen, 1997; Reynolds et al., 1999). After completion of the initial phytoremediation

project in 1996, the site was not actively managed for approximately 15 years, with no nutrient or re-seeding amendments made.

**Table 1. Soil TPH measurements from each plot. For each plot three subsections were measured, data presented as average and standard error. Soil TPH measurements are presented for three time points: after one month of**
**phytoremediation treatment (1995), one year of treatment (1996) and 16 years of treatment (2011). Data reaggregated from (Reynolds et al., 1997b, a; Leewis et al., 2013)**

| Contaminant | Original Treatment | TPH (ppm) 1995 | | | TPH (ppm) 1996 | | | TPH (ppm) 2011 | | |
|---|---|---|---|---|---|---|---|---|---|---|
| CrudeOil | c1 | 5313.95 | ± | 138.03 | 3948.04 | ± | 688.59 | 656.00 | ± | 21.67 |
| | c2 | 5562.93 | ± | 625.89 | 4409.48 | ± | 44.41 | 744.67 | ± | 53.04 |
| | f | 4772.65 | ± | 220.63 | 4017.23 | ± | 372.17 | 631.67 | ± | 11.46 |
| | p1 | 5897.50 | ± | 648.80 | 3879.16 | ± | 169.99 | 664.67 | ± | 38.00 |
| | p1f | 4806.89 | ± | 209.04 | 3374.28 | ± | 545.00 | 638.33 | ± | 37.29 |
| | p2 | 6216.97 | ± | 1369.67 | 3849.75 | ± | 152.26 | 759.00 | ± | 18.25 |
| | p2f | 4067.25 | ± | 893.86 | 3191.01 | ± | 269.57 | 737.67 | ± | 22.41 |
| Diesel | c1 | 5652.96 | ± | 181.57 | 2904.77 | ± | 386.75 | 316.00 | ± | 15.56 |
| | c2 | 4919.41 | ± | 1165.74 | 2105.60 | ± | 262.69 | 339.33 | ± | 7.43 |
| | f | 3538.46 | ± | 1057.88 | 1313.56 | ± | 108.73 | 403.33 | ± | 11.86 |
| | p1 | 3280.68 | ± | 820.33 | 2951.23 | ± | 510.57 | 340.67 | ± | 16.69 |
| | p1f | 3103.13 | ± | 2203.77 | 1192.71 | ± | 247.06 | 430.33 | ± | 18.59 |
| | p2 | 4267.83 | ± | 140.19 | 2746.77 | ± | 158.34 | 392.33 | ± | 15.83 |
| | p2f | 1998.09 | ± | 386.71 | 1071.39 | ± | 380.47 | 395.67 | ± | 9.86 |

**2.2 Sample Collection and Vegetation Characterization**

In June 2011, we initiated a follow-up study of the initial phytoremediation experiment to determine the continuing
influence of original treatment on current TPH concentrations, microbial community presence and abundance, and plant community composition. Due to the lack of replication in the original study design, we divided each of the original treatment



plots into six 1 x 1.5 m² "subsections" (Fig. 1C). From each subsection, we collected soil samples into both sterile plastic zip-top bags for microbial analyses and a glass jar with Teflon lined lids for TPH analysis. Samples were sieved through a sterile 2.5 mm sieve and stored at A) -80 °C for molecular and TPH analysis or, B) at 4 °C for culture-based microbial assays (i.e.,

most-probable number analyses - MPN) which took place within one week of collection. Due to the number of subsamples and to reduce the overall costs of analyses, we conducted analyses on the following number of subsections per plot: A) culturable diesel degrading microorganisms (DDM) – six subsections, B) PLFAs – four subsections, C) bacterial community structure and TPH – three subsections. Reported values are averages with standard deviations.

At the time of soil sampling, we also photographed each individual subsection to estimate percent cover of vegetation.
Vegetation coverage was classified into four "types": grasses (including live and dead), forbs (herbaceous flowering plants; for example, fireweed, dandelions, vetch, Oxytropus, etc.), trees, and bare ground (including crusts and moss cover). Each photo was overlain with a 5 x 5 grid, and if an individual block within the grid was covered more than 50% with a vegetation type, then that block was counted as positive for that vegetation type (Vanha-Majamaa et al., 2000; Chen et al., 2010).

In Leewis *et al*., 2013 we described the soil chemistry, plant community composition, TPH concentrations, and
bacterial community as determined by a molecular fingerprinting technique (i.e., terminal restriction fragment length polymorphism - T-RFLP). Here, we report additional analyses of the same soil samples to further characterize the microbial community using: A) culture-based techniques to determine the ability of the soil community to use TPH as a carbon source (MPNs), B) phospholipid fatty acids (PLFAs) to fingerprint the entire soil community (including bacteria, fungi, and protozoa), and C) 16S rRNA gene sequencing to determine the identity of bulk soil bacterial community members. We additionally
incorporated the newly analyzed plant percent cover data, and previously reported soil chemistry, TPH, and plant community count data into statistical analyses of the microbial community, as detailed below.

**2.3 Phospholipid Fatty Acid (PLFA) Assay**

PLFA extractions and analyses were performed following the methods of (Wu et al., 2009) with minor modifications as follows. Each soil sample (ca. 4.8 g) was freeze-dried and the weight after freeze-drying was recorded for data analysis. All
drying steps were performed with $N_2$ at a temperature below 40 °C. Phospholipids were extracted using 8 mL methanol (HPLC grade, Fischer Scientific, Pittsburgh, PA). Fatty acid methyl esters were extracted using hexane (3 x 1 mL) and then were dried

under N$_2$ (HLPC-grade hexane, EM, (Germany). Prior to GC analysis, 80 μL of the internal standard, methyl nonadecanoate (0.737 nmol/μL; Sigma-Aldrich, Switzerland), was added and the samples were dried completely. The samples were then dissolved in 80 μL hexane and analyzed on an Agilent 6890 N Gas Chromatograph, using an Agilent 19091B-102 (25.0m x

200μm x 0.33μm) capillary column, and MIDI peak identification software using an internal standard (Version, 6.1; MIDI Inc., Newark, DE). PLFA biomarkers and biomass were assigned to microbial groupings as outlined in Table S1 (Song et al., 2008; Kaiser et al., 2010; Ngosong et al., 2012; Stella et al., 2015; Oburger et al., 2016; Polivkova et al., 2018).

## 2.4 Enumeration of Diesel-Degrading Microorganisms

The abundance of culturable aerobic DDM was determined by a 96-well plate most-probable-number (MPN) method

adapted from (Haines et al., 1996), following the modifications detailed in Leewis *et al.*, (2016a). Soils were stored at 4 °C and processed within a week of collection.

## 2.5 Amplicon Preparation and Sequencing

Soil DNA was extracted from a 0.5 g soil subsample using the FastDNA SPIN kit for soil (MP Biomedicals, Ohio, USA) following the manufacturer's instructions. DNA was eluted into 50 μL of water and stored at -20 °C until analysis. DNA

concentrations were evaluated by measuring absorbance at 260 and 280 nm using a NanoDrop ND-1000 Spectrophotometer (Thermo Fischer Scientific, USA). Region V4-V5 of the 16S rRNA gene were amplified using primers 563-577F, 5'-AYTGGGYDTAAAGNG-3', and 926-909R, 5'-CCGTCAATTCMTTTRAGT-3' (Uhlik et al., 2012; Cole et al., 2009). Each of the primers was synthesized with sequencing adaptors (454 Sequencing Application Brief No. 001-2009, Roche), the forward primer was also modified with different barcodes (454 Sequencing Technical Bulletin No. 005-2009, Roche) to allow

for multiple samples to be pooled for sequencing. The PCR mixture (final volume, 25 μl) contained 1 μl each primer (10 μM), 0.5 μl dNTP mix (10 mM), 2.5 μl FastStart 10 X Buffer #2, 0.25 μl FastStart HiFi Polymerase (5 U/μl), and 18.75 μl molecular biology grade water (Roche). The following thermal cycling scheme was used: initial denaturation at 95 °C for 3 min and 30 cycles of denaturation at 95 °C for 30 sec, annealing for 1 min at 55 °C, and extension at 72 °C for 1.5 min, followed by a final extension period at 72 °C for 10 min. Each PCR product was obtained in three parallel reactions, the resulting preparations

were mixed, purified using Pure-Link PCR purification kit (Invitrogen, USA) and pooled for downstream sequencing. Roche

454 GS FLX Titanium sequencing (454 Life Sciences, USA) was performed on pooled reactions at the Utah State University

Center for Integrated Biosystems, Logan, Utah, USA.

## 2.6 Sequence Analyses

Raw pyrosequencing data (*.sff files) were processed using the mothur software package version 1.45.3 and following

the standardized operating procedure (Schloss et al., 2011), accessed July 2021, with minor modifications as described

previously (Uhlik et al., 2012). Sequences with > 3% sequence similarity were grouped into the same operational taxonomic

units (OTUs). These were then classified by mothur-implemented RDP reference files (trainset18_062020; accessed July 2021)

(Wang et al., 2007; Cole et al., 2014) and sequences associated with mitochondria, chloroplasts, archaea, eukaryote, and

unknown Kingdoms were removed. Data were then rarefied to the minimum library size of 3537 sequences. Pyrosequencing

reads were deposited in NCBI Short Read Archive under bioproject number PRJNA950456.

## 2.7 Statistical Analyses

All statistical analyses were performed in R version 3.6.1 (R Core, 2018). To determine difference in the means of

variables, we used a Kruskal-Wallace (KW) rank sum test with original treatment as the factors. We then used a pairwise

Wilcox test with no assumption of equal variance and corrected for multiple comparisons using a Benjamini–Hochberg (BH)

correction to further test differences between individual original treatments. Pearsons's correlations (r) were used to examine

relationships between independent variables.

Alpha and beta diversity metrics were calculated using the phyloseq package (McMurdie and Holmes, 2013) on

rarefied bacterial sequence data and tested using the vegan package (Oksanen et al., 2017). The influence of treatment and soil

properties on bacterial communities was visualized using non-metric multidimensional scaling (NMDS, *metaMDS()*) with

vector fitting of associated soil and plant data (*envfit()*) and tested using a permutational multivariate analysis of variance

(PERMANOVA, *adonis()*). Strongly autocorrelated soil chemical data was removed from the vector fitting prior to analysis

(*cor()*, Pearson's r > 0.70), P-values were corrected using the *p.adjust* function (*method = "BH"*). The relationship between

microbial community structure and measured soil properties was assessed using a Mantel test (*mantel()*). Multi-response

permutation procedure (MRPP, *mrpp()*) was used to test the influence of original treatment or soil type on microbial community

structure. Multivariate testing of the microbial community was conducted using the Bray-Curtis dissimilarity measure
(*distance = "bray"*) and significance quantified using permutations tests (*permutations = 999*). Major OTUs representing
more than 2% of the total communities were depicted in a heat map using the *heatmap2* function.

## 3. Results

### 3.1 TPH and Soil Chemistry

The results detailing soil chemistry and TPH concentrations are described in detail in (Leewis et al., 2013); we briefly
summarize those findings here (Table 1 and S2). The initial TPH concentrations measured in crude-oil-contaminated soils was
6070 ppm, and 8350 ppm in diesel-contaminated soils (Reynolds et al., 1997b, a).  In 1996, after almost a year of treatment,
crude-oil-contaminated soils were 27% - 47% lower than initially measured concentrations. In diesel-contaminated soils, soil
TPH concentrations were 65% - 87% lower than initial measurements (Table 1). In 2011, soil TPH concentrations were 80 -

95% lower than the concentrations measured in 1996. Concentrations of remaining TPH were higher in crude-oil-contaminated
soils (ca. 631 – 760 ppm) compared to diesel-contaminated soils (ca. 319 – 430 ppm). Differences in lingering TPH
concentrations may be associated with the original soil contaminant, as crude-oil contains a more recalcitrant compounds than
refined diesel fuels, or the soil quality itself, as the crude-oil-contaminated soils were coarser in texture more nutrient poor
compared to diesel-contaminated soils (Table S2).

Unfortunately, the initial 1995-1996 investigation did not include measurements of soil chemistry or texture; those data are
only available for soils collected in 2011 (Table S2). Overall, diesel-contaminated soils were significantly higher in total
carbon, NO3-N, and K, with finer soil texture than crude-oil contaminated soils (KW P < 0.01). Soils which had been originally
fertilized remained significantly higher in total P and K compared to unfertilized soils (KW P < 0.001), regardless of
contaminant type.

**3.2 Long-term Responses of Plant Community to Phytoremediation Treatment**

       After 15 years with no active site management, the plant community within each treatment plot changed significantly
from the first establishment of the site in 1995, and none of the originally planted grasses were present at the time of sample
collection (Leewis et al., 2013). Plant communities differed according to soil type (MRPP A = 0.175, *P* = 0.001): in the finer-



textured diesel-contaminated soils, there were more, and larger, woody plants (e.g., *Salix* spp., *Populus* spp., *Betula*

*neoalaskana*, and *Picea glauca*) present when compared to the crude oil-contaminated soils (Table S3). The crude oil plots,

with coarser-textured soils, were more heavily colonized by non-native plants from the family *Fabaceae* such as *Taraxacum*

*officinale* (dandelion), *Vicia cracca* (common vetch), and *Trifolium hydridum* (clover). Native plants including *Ledum*

*decumbens* (Labrador Tea), *Epilobium angustifolium* (fireweed), along with willow (*Salix* spp.), spruce (*Picea* spp.), and

poplar seedlings (*Populus* spp.) were present throughout the site, regardless of soil type (i.e., crude oil-contaminated coarse-

textured soils or diesel-contaminated fine-textured soils). In both soils, plots originally fertilized had greater coverage with

grasses and less bare ground than other plots (Fig. 2). Plots not originally fertilized had more coverage of trees compared to

other plots (Fig. 2). Original phytoremediation treatment explained 63% of the variation in the current plant community

structure (PERMANOVA CO: $P = 0.001$, $R^2 = 0.630$, F = 3.978; DE: $P = 0.001$, $R^2 = 0.635$, F = 4.077).

As previously reported, we found that increased TPH disappearance was associated with increased numbers of native

trees and shrubs such as willow (*Salix* spp.), birch (*B. neoalaskana*), white spruce (*S. glauca*), and balsam poplar (*Populus*

*balsamifera*), regardless of the original phytoremediation treatment (Leewis et al., 2013).





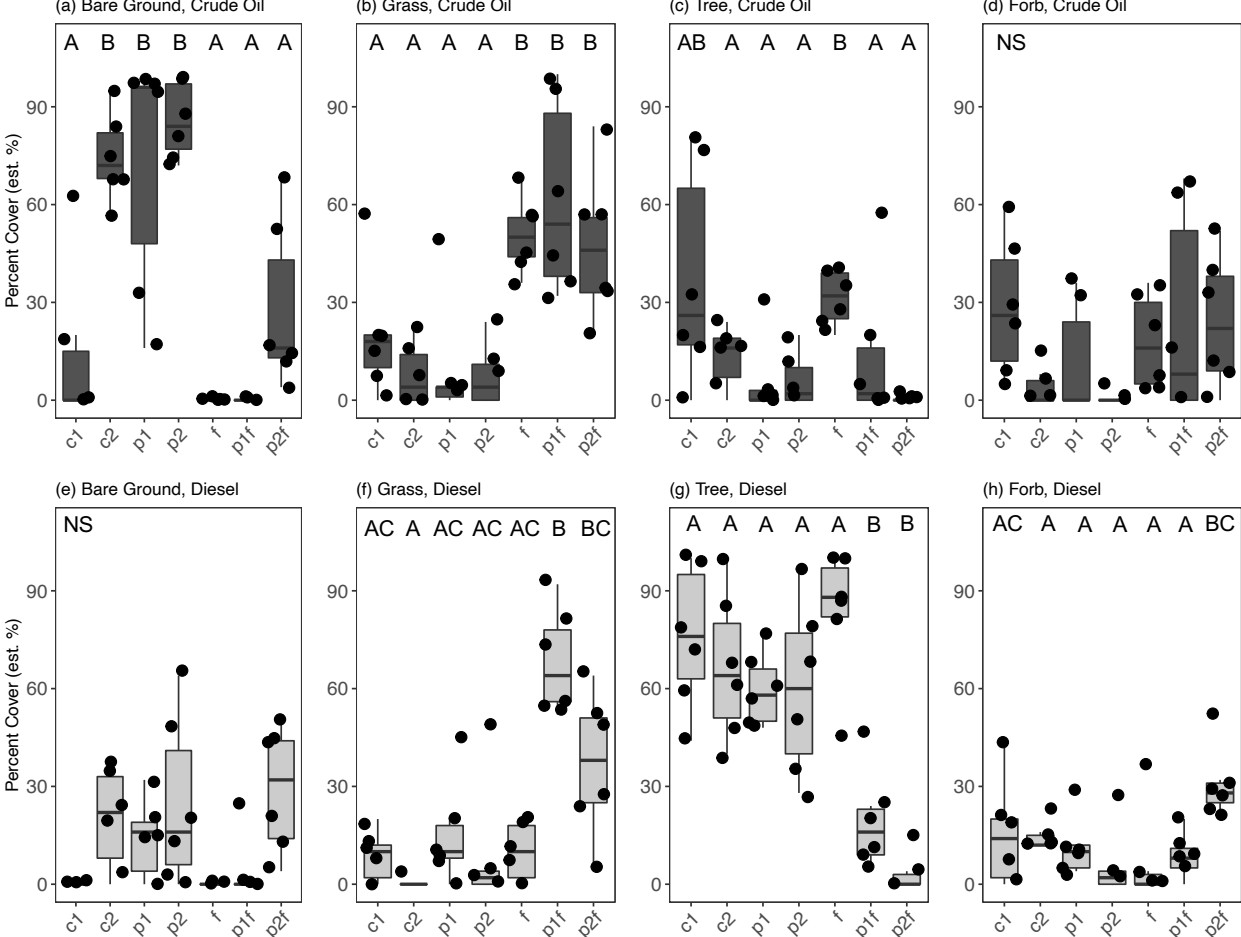

Figure 2: **Estimated percent cover of vegetation groups in either Crude Oil (dark grey) or Diesel (light grey) plots.**
**Measurements were based on visual estimates in each of the six sub-plots. The values shown are means with 95%**
**confidence intervals (N = 6), note y-axes have different scales. Treatments indicated are as follows: no treatment (c1,**
**c2), planted with annual ryegrass (p1), a mix of annual ryegrass and Arctared fescue (p2), treated with fertilizer (f),**
**and/or no added nutrients (no "f" indicated). Significant differences in percent cover are indicated by different letters,**
**"NS" indicates no significant differences found.**



**3.3 Quantity of Diesel-Degrading Microorganisms is associated with TPH, Vegetation**

To assess the ability of the soil community to use PHC as a carbon source, we measured the MPN of culturable DDMs

across all the treatment plots. Overall, there were significantly more DDM in the crude-oil-contaminated, gravely soil (average

$5.57 \pm 0.71$ log MPN / g soil) than diesel-contaminated soils ($4.31 \pm 0.77$ log MPN / g soil; $P < 0.05$ CO vs DE; Fig. S1). We

also found differences in the amount of DDM associated with the original phytoremediation treatment: in the crude-oil soils,

an initially untreated plot ($6.34 \pm 0.24$ log MPN / g soil) and the fertilizer-only plot ($6.43 \pm 0.61$ log MPN / g soil) had the

highest numbers of DDM ($P < 0.04$, Kruskal-Wallis). In diesel soils, one initially untreated plot had significantly higher

numbers of DDM ($5.35 \pm 0.81$ log MPN / g soil) than the other treated plots ($P < 0.04$, Kruskal-Wallis).

We next examined the relationship between DDM and other factors on the site at the time of collection, including

TPH concentrations and current vegetation. There was a weak inverse relationship between counts of DDM and TPH

concentrations in both soil types in which increased counts correlated to decreased TPH concentrations, although this

relationship was less strong in crude-oil soils (Pearson's r = -0.297, $P = 0.191$, Fig. 3A) than in the diesel-contaminated soils

(Pearson's r = -0.387, $P = 0.083$, Fig. 3B). In both crude-oil and diesel soils, there was a positive relationship between DDM

and the number of plants on the site, although this relationship was only significant in crude oil-contaminated soils (Pearson's

r = 0.362, $P = 0.019$). We further explored this plant-contaminant relationship and found a significant positive relationship

between DDM and the percent cover of trees ($P = 0.005$, Pearson's r = 0.586) in the crude-oil plots (Fig. 3C), but not in the

diesel plots (Fig. 3D).




**Figure 3: Relationship between the number of culturable diesel-degrading microorganisms (DDM) and total petroleum**
**hydrocarbon concentrations (TPH; A & B) or percent cover of trees (C & D) in soils contaminated with crude oil (A &**
**C) or diesel (B & D). Solid symbols indicate treatments originally fertilized. Treatments indicated are as follows: no**
**treatment (c1, c2), planted with annual ryegrass (p1), a mix of annual ryegrass and Arctared fescue (p2), treated with**
**fertilizer (f), and/or no added nutrients (no "f" indicated).**

**3.4 Microbial Biomass is Influenced by Contaminant and Treatment**

Both the original phytoremediation treatment and soil type (i.e. crude oil-contaminated coarse-textured soils or diesel-
contaminated fine-textured soils) significantly influenced microbial biomass, as indicated by analysis of PLFAs. Total

microbial biomass was generally higher in plots which had been fertilized compared to plots that had not received original fertilization (Table S4 & Fig. S2, P < 0.01, Kruskal-Wallis). The same held true for all other individual PLFA biomarkers (P < 0.05, Kruskal-Wallis) in both crude-oil and diesel plots. The one exception was in the crude oil plots, where PLFAs associated with Actinobacteria were not significantly lower in originally fertilized plots ($P = 0.188$ Kruskal-Wallis; Table S4 & Fig. S2). Plots originally unplanted and unfertilized ("control"), or plots originally planted only with ryegrass ("planted", P < 0.02, Kruskal-Wallis) had the lowest overall microbial biomass in both crude oil and diesel soils.

### 3.5 Soil Microbial Communities are Influenced by Treatment and Vegetation

We further tested the influence of original treatment and soil properties on microbial community structure as defined by PLFA biomarkers using PERMANOVA. Most variation in the composition of PFLA biomarkers was explained by original phytoremediation treatment (CO contaminated soils: 82%, DE contaminated soils 61%) and by the presence of fertilizer in that treatment (CO contaminated soils: 36%, DE contaminated soils 24%; Table 2). In diesel soils, the concentration of TPH and $NO_3^-$ explained ca. 20% variation in the composition of PLFA biomarkers (Table 2). In crude oil soils, other measured soil properties, such as percent C, pH, $NO_3^-$, K, P, and CEC, also significantly explained variation in the composition of PFLA biomarkers (Table 2). In crude oil-contaminated soils coverage of grasses, amount of bare ground, and (marginally) coverage of forbs were significantly associated with variation in the composition of PLFA biomarkers (Table 2).

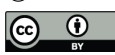
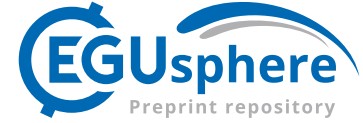

**Table 2. Association of soil microbial community structure, as measured by PLFA biomarkers or 16S rRNA gene sequencing, with factors such as original treatment type, presence of fertilizer in original treatment, amount of vegetation present on the site at the time of sample collection, or soil nutrient and structure data. P values adjusted for multiple comparisons using BH.**


| Factor | Crude Oil | | | | | | Diesel | | | | | |
|---|---|---|---|---|---|---|---|---|---|---|---|---|
| | PLFA | | | 16S rRNA | | | PLFA | | | 16S rRNA | | |
| | R2 | P (adj) | F | R2 | P (adj) | F | R2 | P (adj) | F | R2 | P (adj) | F |
| Original Phytoremediation | | | | | | | | | | | | |
| Treatment | 0.82 | **0.0037** | 10.46 | 0.64 | **0.0037** | 4.14 | 0.61 | **0.01792** | 3.59 | 0.58 | **0.004** | 3.23 |
| Fertilization | 0.36 | **0.0059** | 10.48 | 0.30 | **0.0037** | 8.07 | 0.24 | **0.0328** | 6.07 | 0.22 | **0.0037** | 5.31 |
| Tree (% cover) | 0.12 | 0.1472 | 2.67 | 0.10 | *0.0747* | 2.1634 | 0.05 | 0.3264 | 1.10 | 0.11 | **0.0160** | 2.46 |
| Grass (% cover) | 0.32 | **0.008** | 8.88 | 0.24 | **0.0053** | 5.9332 | 0.02 | 0.6330 | 0.39 | 0.12 | **0.0213** | 2.51 |
| Forb (% cover) | 0.20 | *0.057* | 4.70 | 0.11 | *0.0747* | 2.3772 | 0.08 | 0.2400 | 1.69 | 0.07 | 0.2240 | 1.39 |
| Bare Ground (% cover) | 0.62 | **0.0053** | 31.64 | 0.30 | **0.0053** | 8.0259 | 0.09 | 0.2400 | 1.80 | 0.06 | 0.3264 | 1.15 |
| TPH | 0.07 | 0.2987 | 1.36 | 0.05 | 0.3837 | 1.08 | 0.21 | **0.0470** | 4.98 | 0.11 | **0.0215** | 2.33 |
| pH | 0.57 | **0.0037** | 25.07 | 0.31 | **0.0037** | 8.57 | 0.19 | *0.0645* | 4.39 | 0.19 | **0.004** | 4.45 |
| NO3- | 0.35 | **0.0059** | 10.56 | 0.16 | **0.016** | 3.63 | 0.27 | **0.01704** | 7.05 | 0.07 | 0.179 | 1.46 |
| P | 0.28 | **0.026** | 7.55 | 0.25 | **0.0037** | 6.39 | 0.08 | 0.26437 | 1.63 | 0.20 | **0.006** | 4.77 |
| K | 0.53 | **0.0059** | 21.46 | 0.31 | **0.0037** | 8.34 | 0.02 | 0.10095 | 0.39 | 0.16 | **0.004** | 3.51 |
| CEC | 0.51 | **0.0037** | 20.00 | 0.22 | **0.0037** | 5.35 | 0.11 | 0.168 | 2.43 | 0.11 | **0.026** | 2.24 |
| %C | 0.52 | **0.0037** | 20.98 | 0.25 | **0.0037** | 6.40 | 0.12 | 0.12821 | 2.64 | 0.07 | 0.176 | 1.46 |
| Sand | 0.07 | 0.3007 | 1.33 | 0.04 | 0.6623 | 0.77 | 0.02 | 0.748 | 0.32 | 0.10 | **0.037** | 2.11 |
| Silt | 0.04 | 0.4947 | 0.78 | 0.04 | 0.7249 | 0.69 | 0.02 | 0.70158 | 0.41 | 0.12 | **0.017** | 2.56 |
| Clay | 0.04 | 0.4995 | 0.78 | 0.11 | *0.064* | 2.31 | 0.09 | 0.192 | 1.98 | 0.11 | **0.008** | 2.31 |





Ordination of PLFA biomarkers indicated that soil community structure was associated with original

phytoremediation treatment type (Fig. 4A/B). We tested the strength of these associations and found the relationship between

original treatment and PLFA-derived biomarkers was stronger in the crude-oil-contaminated soils (MRPP A = 0.542, $P$ =

0.001) than the diesel-contaminated soils (MRPP A = 0.268, $P$ = 0.005). A vector-fitting method used to interpret the

ordinations using measured soil chemical properties indicated that in the crude-oil soils several factors were significantly

associated with the distribution of PLFA biomarkers in ordination space, particularly in plots that were initially fertilized which

were associated with increase soil nutrients (e.g., P, K) and other metrics of soil health (e.g., near neutral pH, CEC, and %C;

Fig. 4A). In diesel-contaminated soils however, only $NO_3^-$ was associated with the distribution of PLFA biomarkers in

ordination space (Fig. 4B).





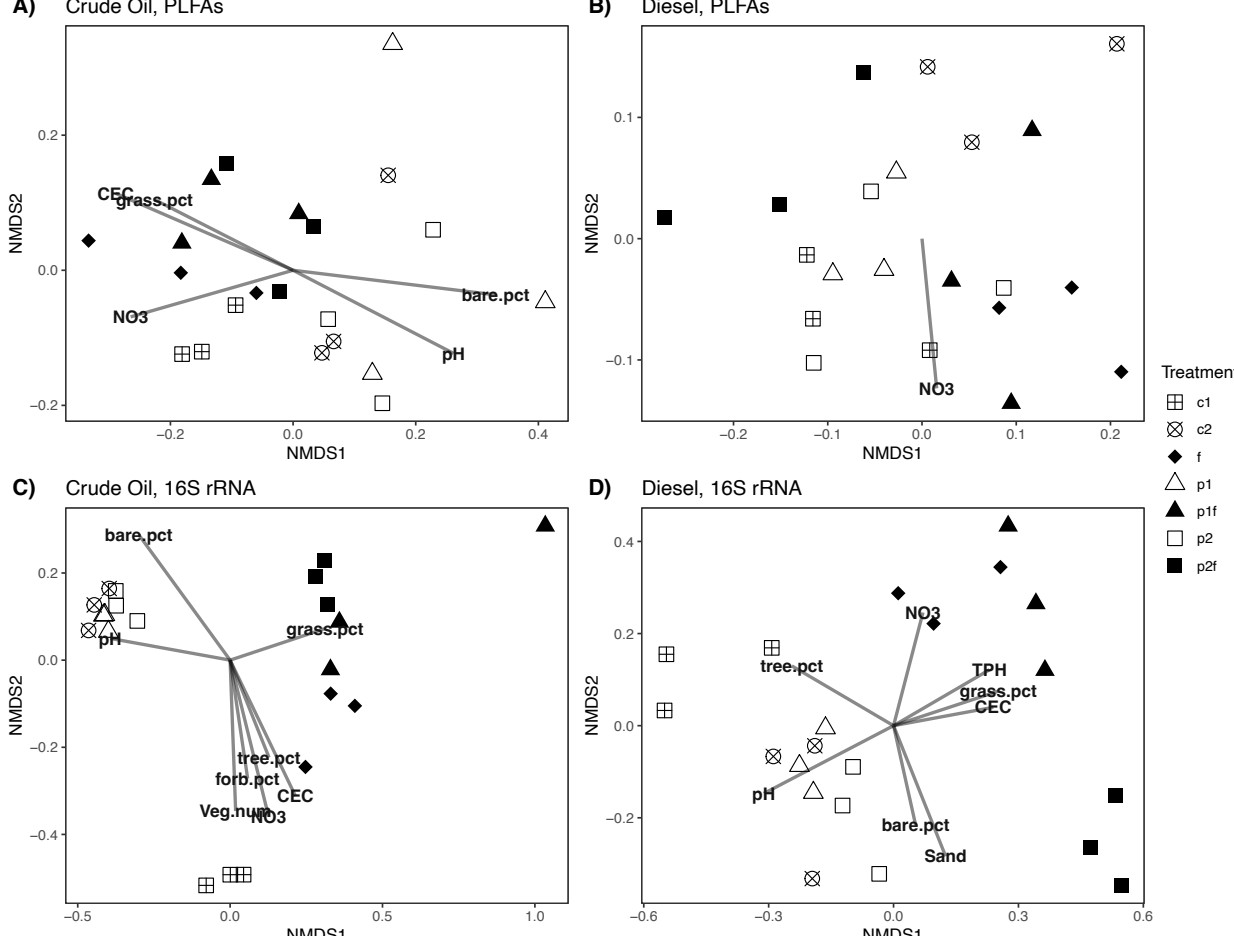

**Figure 4. Non-metric multidimensional scaling ordination analysis (NMDS) of soil PLFAs (A & B) or 16S rRNA genes (C & D) from crude oil (A, C) or diesel (B, D) contaminated soils, with subsequent fitting of environmental vectors onto the ordination (P < 0.05). The vectors are as follows: NO3 = nitrate, CEC = cation exchange coefficient, Veg.num =total vegetation counts (excluding trees), Tree.num = total number of trees, TPH = total petroleum hydrocarbons, Sand = soil texture (i.e., %sand, % silt, % clay). Solid symbols indicate treatments originally fertilized. Treatments indicated are as follows: no treatment (c1, c2), planted with annual ryegrass (p1), a mix of annual ryegrass and Arctared fescue (p2), treated with fertilizer (f), and/or no added nutrients (no "f" indicated).**

Analysis of bacterial community structure using 16S rRNA gene sequencing revealed the significant influence of both soil type (diesel-contaminated silty soil vs crude oil-contaminated gravelly soil) and original phytoremediation treatment (Table S5). Alpha diversity of the bacterial community as measured by the Shannon Index was higher in the crude-oil soils





when compared to the diesel soils ($P$ = 0.0004, Kruskal-Wallis). In crude-oil soils, there was a significant effect of phytoremediation treatment, and alpha diversity was higher in plots that had been originally fertilized ($P$ = 0.022, Kruskal-Wallis). There was no observed effect of original phytoremediation treatment observed on alpha diversity in diesel contaminated soils (P > 0.1, Kruskal-Wallis).

To determine differences among bacterial communities, we constructed a heatmap of families which were at least 2% abundant in more than one sample (Fig. S3). The family *Bradyrhizobiaceae* dominated the bacterial community in diesel-contaminated soils (27.62%) and was also abundant in the crude-oil soils (8.89%). The family *Acidobacteria Gp6* was also abundant across all the treatments and contaminant types, with a higher relative abundance in crude-oil soils (11.60%) than in diesel soils (8.36%). Members of the families *Gemmatimonadaceae* (crude oil: 4.20%, diesel: 7.63%), *Rhodospirillaceae*

(crude oil: 1.73%, diesel: 2.66%), *Xanthomonadaceae* (crude oil: 1.36%, diesel: 1.08%), and *Caulobacteraceae* (crude oil: 1.40%, diesel: 1.69%), among others, were present in all investigated soils (Fig. S3). At all sites, there was also a large percentage of bacteria unclassified at the family level (diesel, 9.81%; crude oil, 13.29%), suggesting that they may be novel sequences of uncharacterized bacteria (Fig. S3).

        To assess differences among bacterial community structure in each phytoremediation treatment plot, we used an

NMDS plot. Bacterial communities were significantly associated with original phytoremediation treatment; plots that were originally fertilized grouped separately from other treatments (Fig. 4C & D; CO MRPP A = 0.214, $P$ = 0.013; DE MRPP A = 0.356, $P$ = 0.002). Vector fitting analysis indicated that pH, soil chemical properties (e.g., $NO_3^-$, CEC, nutrients) heavily influenced the bacterial community structure in both soil types (Fig. 4). In both soil types, the number of trees present on each plot was associated with bacterial community structure in the originally unplanted or unfertilized plots.

We further examined the relationship between soil and phytoremediation treatment factors with bacterial community structure using a PERMANOVA test. In both soil types, original treatment type explained more than half of the variation in bacterial community structure (CO 64%, DE 58%) and the presence of fertilizer in the original treatment explained approximately a quarter of the variation (CO 30%, DE 22%). Measures of soil health, such as near-neutral pH, nutrient concentrations (P, K), CEC, and soil texture, were also associated with the variation in bacterial community structure in both

crude oil and diesel-contaminated soils (Table 2). The plant communities present also helped explain variation in bacterial

community structure. In diesel-contaminated soils, percent coverage of trees and grasses was associated with bacterial

community structure (Table 2). In crude oil-contaminated soils, coverage of grasses, amount of bare ground, and (marginally)

coverage of forbs were significantly associated with bacterial community structure (Table 2). We then used a Mantel test to

further explore these relationships between the species composition of plant communities and the bacterial community

structure. There was a strong relationship between the bacterial community and the vegetation present on the site in both crude

oil (Mantel r = 0.2147, $P$ = 0.013) and diesel-contaminated soils (Mantel r = 0.3568, $P$ = 0.002).

## 4 Discussion

Here, we investigated the long-term influence of phytoremediation on recruited plant communities, contaminant

disappearance, and soil microbial community structure and capacity for PHC degradation, 15 years after phytoremediation

treatment activities had stopped. We found that phytoremediation treatment had deterministic effects on the recruited

vegetation and associated bulk soil microbial communities (Table 2, Fig. 2, Fig. 4), which in turn influenced the continuing

contaminant disappearance in both crude-oil and diesel-contaminated soils (Table 1, Fig. S1, Fig. 3). Together with the

measurements of soil microbial community structure and PHC degradation capacity, the vegetation and TPH data provide a

more holistic, long-term view of the impact of treatment on plants and soil microbial communities at this site.

Long after active phytoremediation ended, variation in plant communities across the site was most strongly shaped

by the original treatment, although the study design did not allow us to fully determine the cause-effect nature of this

relationship (Table 2, Fig. 2). In particular, the initial use of fertilizer was strongly associated with more vegetation coverage,

whereas plots without fertilizer application had more exposed bare ground. While the direct effects of fertilizer application

may have been relatively transient, on the order of months, we did find that soil initially fertilized continued to have higher

total P and K than other soils (Table S2).  It is possible that the original fertilization acted to allow seeds to colonize and

establish in treated plots thereby priming plant succession. Many early successional plant species such as forbs, grasses and

deciduous trees produce litter that is of higher quality and therefore decomposes more rapidly than secondary successional

species such as coniferous trees (Allison and Treseder, 2011). The subsequent turnover of this nutrient-rich plant biomass may

also have played a role in setting plant communities on different successional trajectories through longer-term enhancement



of available nitrogen (Clark et al., 2007; Cleland and Harpole, 2010). Indeed, other studies in early successional boreal forests have shown that fertilization can shift the plant community toward species with higher-quality litter and increased litter decay rates compared to unfertilized plots (Ruess et al., 1996; Allison et al., 2010).

The continuing influence of the original phytoremediation treatment was also evident in the soil microbial community (Table 2, Fig. 4, Fig. S3). In particular, the initial use of fertilizer had sustained effects on the bulk soil microbial community,

accounting for differences in both total microbial biomass and bacterial community structure. Microbial biomass was higher in soils initially fertilized, and the total microbial communities as measured by PLFAs and 16S rRNA amplicon sequencing were structured according to the original treatment and the use of fertilizer. This is in agreement with several long-term studies which have found that initial fertilization in unmanaged systems can lead to sustained differences in microbial communities (Talbot and Treseder, 2012; Geisseler and Scow, 2014; Lupwayi et al., 2021). These treatment-scale effects can extend beyond

the bulk-soil community into the endosphere of the plants themselves. In a sister study, we also found that the choice of initial phytoremediation strategy drove the succession of endophytic bacteria associated with colonizing vegetation (Papik et al., 2023).

Soil physiochemical properties such as pH and texture are known to influence microbial communities (Sutton et al., 2013; Leewis et al., 2022). The two soils used in our study differed substantially including different original contaminant,

origin, texture, pH, and nutrient status (Table 1). In our previous study of this site, (Leewis et al. 2013), we reported that these dissimilarities led to distinctions in the plants recruited on the two soils, with more primary successional and invasive plants present on the coarse-textured crude oil contaminated soils. Here, we see how the soil physiochemical effects also influenced the microbial communities and their response to the initial treatments; the bacterial community in gravelly crude-oil soils had a higher diversity than the finer-textured diesel soils. Total soil microbial communities and the bacterial community were all

more strongly associated with initial phytoremediation treatment and fertilization in the coarse crude-oil-contaminated soils than those in the diesel-contaminated soils (Table 2). Coarse-textured soils tend to have less water and nutrient retention, conditions which are less hospitable for plant and microbial growth. Treatment such as fertilization and planting can lead to major changes in soil physiochemical properties, such as increasing soil organic matter content and nutrient retention, which



can, in turn, impact the soil community (Reeve et al., 2010; Blanchet et al., 2016). Therefore, the lingering effect of treatment

would be more evident in soils that started out as harsher for microbial and plant growth.

Original phytoremediation treatment strategy influenced the vegetation recruited, which in turn influenced microbial community structure. While it is well known that both plants and fertilizer can cause shifts in soil microbial community structure, here it is almost impossible to tease apart their separate contributions to the microbial community because the vegetation on each plot is driven by the initial phytoremediation treatment. Our results indicate that the percent coverage of

vegetation, rather than the individual plant species themselves, was an important factor in driving soil community structure. Similarly (Brown and Jumpponen, 2014) found that the influence of vegetation composition on bacterial community structure was minor compared to the presence or absence of plants. We also found higher microbial biomass in initially fertilized plots, which may be a consequence of increased vegetation cover. In addition to shifting plant recruitment patterns, fertilization can increase plant biomass, which in turn can increase litter deposition and below-ground organic carbon deposition (Geisseler

and Scow, 2014), and these plant inputs to soil may be a particularly important influence in degraded soil (Knelman et al., 2012).

Microbial community function is also known to be influenced by the presence and diversity of plants (Leewis et al., 2016; Musilova et al., 2016; Schmid et al., 2021). Here we found that the number of culturable microorganisms able to degrade TPH was most strongly associated with the plant communities present on the site. In our previous investigation of the site, we

found an inverse relationship between increasing counts of woody vegetation and decreasing TPH concentrations (Leewis et al., 2013). This further investigation found increasing DDM with more tree coverage and increasing DDM associated with decreasing TPH concentrations (Fig. S1 & Fig. 3). This indicates that microbial biodegradation of PHCs may be an on-going process, despite very low TPH concentrations, and that the microbial functional potential is stimulated by, or at least associated with, the current vegetation on site. It has been hypothesized that plants with higher amounts of some secondary plant

metabolites, such as aromatics, may lead to increased biodegradation potential in contaminated soils (Fletcher and Hegde, 1995; Singer et al., 2004; Slater et al., 2011). At higher latitudes, trees have increased concentrations of secondary metabolites when compared to similar tree species growing at lower latitudes (Stark et al., 2008), this includes many species identified on the investigated sites such as willow (*Salix* spp.; Table S2). Our data indicate that the presence and coverage of native Alaskan



tree species are directly related to increased degradation potential and TPH disappearance in both crude-oil- and diesel-

contaminated soils.

        Phytoremediation is a common practice for restoration of sites contaminated with organic pollutants, with most

studies pursuing results over the short- to mid-term range (< 5 years (Palmroth et al., 2002; White et al., 2006; Siciliano et al.,

2003; Elshamy et al., 2019). With phytoremediation treatment, contaminant concentrations can decrease below regulatory

clean-up limits within a short- to mid-term time range in high-latitude soils (Robichaud et al., 2019; Lopez-Echartea et al.,

2020). Even without active treatment, there can be significant decreases in original contaminant concentrations which, over

long timeframes, may result in a site reaching regulatory cleanup limits (Leewis et al., 2013; Robichaud et al., 2019; Lopez-

Echartea et al., 2020). However, accelerating the rate of contaminant disappearance and therefore site remediation, is essential

to increasing the speed of ecosystem recovery and minimizing environmental impacts while still maintaining reasonable costs.

Here we show that minimal initial investment of plants and fertilizer can increase the initial TPH disappearance rates, while

also priming the site for long-term volunteer vegetation cover that supports continued TPH disappearance and ultimately

remediation of the site. While the original grasses planted were associated with significant decreases in soil TPH

concentrations, they required annual replanting. Our results suggest that initial planting with local perennial plants, which are

better adapted to prevailing environmental conditions, may provide a low-cost method to increase long-term phytoremediation

potential (Slater et al., 2011; Robichaud et al., 2019).

**5 Conclusions**

        In conclusion, our study adds to the growing body of knowledge that phytoremediation treatment appears to be an

effective means of reducing petroleum hydrocarbon concentrations in soil and stimulating indigenous TPH-degrading

microorganisms, and that the effects of original treatment can be seen long after active site management has ceased. The data

also indicate the importance of carefully choosing the initial phytoremediation treatments, as these conditions will leave a

lasting legacy on TPH transformation, vegetation recruitment, and soil microbial communities. Our results imply that, through

the use of local perennial vegetation and fertilization, contaminated sites may be primed for long-term and effective

remediation with limited site management.  More research is needed to identify local native vegetation which may be optimal

for initiating plant establishment and microbial succession and accelerating *in situ* remediation processes, particularly in sub-Arctic regions.

**Data availability**

Raw sequence data are available for download from the NCBI Short Read Archive under bioproject number PRJNA950456.

**Author Contributions**

MCL analysed the data & wrote the first draft of the manuscript, with support from OU. MCL and MBL designed the experiment. MCL, MBL, and CK conducted experiments. MBL provided equipment, supplies, and feedback. All authors
provided revisions to the manuscript.

**Conflicts of Interest**

The authors declare that they have no conflict of interest.

**Acknowledgements**

The authors acknowledge and thank C. Michael Reynolds (U.S. Army Corps of Engineers) for earlier site work and
establishment, and Jim Fish (Alaska Department of Environmental Conservation) for introduction to the site. The authors thank Carl Richmond for help with sample collection and processing, and Taylor Gofstein for analytical inspiration.

Research reported in this publication was supported by Institutional Development Awards (IDeA) from the National Institute of General Medical Sciences of the National Institutes of Health under grant number P20GM103395, from the National Centre for Research Resources (NCRR) under grant number 5P20RR016466 and EPSCoR NSF award EPS-0701898.
The content is solely the responsibility of the authors and does not necessarily reflect the official views of National Institutes of Health (NIH). OU acknowledges the support of the INTER-EXCELLENCE program of the Ministry of Education, Youth and Sports of the Czech Republic. Additional financial support was provided by NSF DEB-1257424 (MBL and MCL) and the NSEP David L. Boren Fellowship (MCL).



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
