# Peer review of "Long-term legacy of phytoremediation on plant succession and soil microbial communities in petroleum-contaminated sub-Arctic soils"

_EGUsphere, 2023_

## Author Comment (AC1)

*Response to Reviewer Comment 1* | EGUSPHERE-2023-2097
**"Long-term legacy of phytoremediation on plant succession and soil microbial communities in petroleum-contaminated sub-Arctic soils"**
Mary-Cathrine Leewis, Christopher Kasanke, Ondrej Uhlik, Mary Beth Leigh

*We would like to thank the reviewer for their thoughtful feedback and comments on our manuscript. The reviewer identified several important points that we have addressed which will increase the accessibility and reproducibility of this study. The original comments are written in* black *and our reply in **blue and italics**.*

The manuscript investigates into a reevaluation of a petroleum-contaminated site in Fairbanks, Alaska, fifteen years after the introduction of annual grasses and/or fertilizer treatments to investigate their impact on the degradation of petroleum hydrocarbons. The study meticulously examines contaminant concentrations, as well as the dynamics of plant and soil microbial ecology. This research is particularly significant given the infrequency of long-term phytoremediation studies, acknowledging the necessity for extended investigations in a field where remediation timelines often exceed those of conventional physico-chemical treatments.

The overall quality of the study is commendable, featuring an ample number of figures, tables, and well-referenced content.

I propose the following suggestions to enhance the manuscript:

1. Script Accessibility: It would be beneficial if the authors could provide a link to the scripts employed for statistical and sequencing analyses on a platform such as GitHub. This transparency enhances the reproducibility of the study.

*We appreciate this feedback and agree that it is important to increase the transparency and accessibility of data and associated coding. To this end, we have added the scripts to a github repository (github.com/mcleewis/FarmersLoop) and added the link and coding accessibility statement to the methods section (line 185 and 205, in the revised manuscript).*

2. Pyrosequencing Data Clarification: The authors are encouraged to include a statement elucidating the comparability between results obtained through pyrosequencing and Next-Generation Sequencing (NGS). A justification for the relevance of publishing pyrosequencing data in the context of modern microbiome studies would further strengthen the manuscript.

*We appreciate this feedback that pyrosequencing is an outdated technology however when the original study was conducting, the sequencing was the appropriate technology. While it has*

*taken a long time for this paper to reach review, community assessment by pyrosequencing still has value as a tool. In several studies that explore the two technologies with the same samples, the abundance of genes, genotypes, and communities were found to be highly correlated (R>0.9) (for example: https://journals.plos.org/plosone/article?id=10.1371/journal.pone.0030087 and others). To acknowledge the shortcoming associated with pyrosequencing, we have added a statement to the methods section (line 174-176, in the revised manuscript).*

3. Enhanced Figure Visualization: Figures resembling the color-coded schemes in Figures 3 and 4 would significantly enhance the clarity of the manuscript. The use of color could aid in distinguishing between different treatments, facilitating a more accessible interpretation of the data.

*Thank you for this comment, while the original figures were built in greyscale to ensure that the figures are readable by any with potentially colour blindness, we agree that they benefit from the addition of colours. We have re-made most figures (i.e., Figure 4, 5, S1, S2, S3, S5) to include both colour and changes in shape.*

4. Comparison of Concentrations: While the manuscript notes that crude oil and diesel concentrations remain around 500 ppm after a decade of treatment, it would be valuable to compare and discuss these findings with other bioremediation and phytoremediation studies. An exploration of whether similar concentrations are observed in other studies after a decade of treatment would contribute valuable insights to the field, as those are expected to be lower.

*Thank you for this comment, we have changed the way the TPH data are presented (Table S2) to highlight the overall loss more clearly over the course of all the measured time points. In addition, we added to discussion (line 456-457, in the revised manuscript), to link the percent disappearance presented here to other phytoremediation studies conducted in Northern areas.*

---

## Author Comment (AC2)

*Response to Reviewer Comment 2* | EGUSPHERE-2023-2097
**"Long-term legacy of phytoremediation on plant succession and soil microbial communities in petroleum-contaminated sub-Arctic soils"**
Mary-Cathrine Leewis, Christopher Kasanke, Ondrej Uhlik, Mary Beth Leigh

*We would like to thank the reviewer for their thoughtful critique and comments on our manuscript. The original reviewer comments are written in* black *and our reply in* **blue and italics**.

*The reviewer's comments identified several weaknesses in our presentation of the results that required our attention. In particular, the reviewer makes three main points that are mentioned throughout their review.*

*The first point is the need to clarify what new data is being first presented in this manuscript and to more clearly stat which results had been previously published. We have addressed these comments by expanding the context for this study in the introduction (lines 77-94, in the revised manuscript), as well as detailing the previously published results which provide important context for the current study in a new results section entitled "3.1 Prior Findings on TPH, Soil Properties, and Plant Communities" (lines 207-220, in the revised manuscript).*

*The second point is the need to more clearly present the statistics associated with presented figures and tables. This is an excellent observation, while the statistics were completed it was an oversight to not include all statistics in our initial submission. To this end we have added the results of statistical tests to each table and figure throughout the manuscript using letters to indicate significant differences (P < 0.05).*

*The third point is regarding the microbial sequence data, which the reviewer suggests we conduct additional analyses and provide further discussion of. the presentation and discussion of the microbial data. We thank the reviewer for this suggestion and have included the additional analyses (i.e. differential sequence analysis) and interpretation of the microbial data (lines 365-373, 439-452, in the revised manuscript). We appreciate this suggestion; the additional analyses have added to the overall value of the data presented herein.*

*Finally, the reviewer makes a valid point regarding the breadth of conclusions stated. To this end we have clarified results and focused the discussion away from the more speculative statements (e.g. lines 385-388, line 469-475, in the revised manuscript), to keep within the bounds of the limits presented.*

The manuscript builds on a few prior papers published from a phytoremediation experiment initiated in 1995 in which crude oil- or diesel- contaminated soil received fertilizer, one of two plant treatments, or both to determine whether such treatments impacted the hydrocarbon degradation rate. In 2011 after 15 years without active management, the plant communities for each plot at this site were assessed, and soil samples were taken. A study describing the 2011 plant community at the species level and soil physiochemical properties (including hydrocarbon level) was published in 2013, which also included coarse data on the soil microbial community (T-RFLP analysis).

This current study expands insight into the soil microbial community present in the 2011 samples with: 1) PLFA analysis, to determine total microbial biomass and provide another broad profile of soil microbes; 2) 16S rRNA sequencing, to characterize the diversity of bacterial taxa; and 3) Estimates of the

number of cultivable diesel- degrading microbes present. The plant community data that had already been described in the 2013 paper is also re-aggregated in a coarser way, by % coverage of different vegetative types (grasses, forbs, trees, or bare ground). Soil and plant metrics are linked to microbial data through various analyses.

The methods and analyses are appropriate, though the study is limited by the lack of true replication in the initial experimental design chosen in the 90s. The authors use 454 pyrosequencing which is today an outdated sequencing technology, but was more standard circa 2011 if that is when the initial 16S sequencing was done.

*We agree these are two methodological limitations, which cannot be changed but should be more fully acknowledged. We have added language to address the limitations of the sequencing technology used (lines 174-176, in the revised manuscript), and clarified the limitations of the original study design (lines 117-118, 124-125, in the revised manuscript).*

The long-term nature of the study is commendable, as relatively few papers track the impacts of a single management event after considerable time has passed. There are also some noteworthy patterns in the microbial data. It is striking that bacterial community composition still diverged by initial plant/fertilizer treatment 15 years after the fact, and this contributes to other literature that documents long-term effects of fertilization and/or plant cover on soil microbial community composition. Nonetheless, I feel that further exploration of the 16S data would have strengthened the study. Differential abundance analysis in particular could highlight key OTUs differing between treatments, as was done for rhizosphere and endosphere bacteria of plants harvested from this same experimental site (Papik et al 2023).

*Thank you for this suggestion, as noted above and detailed below we have added the suggested analyses and expanded on our presentation and discussion of the microbial results.*

In general, the focus and clarity of the manuscript could be much improved, particularly concerning which results have been published before and which are presented here for the first time. Moreover, much of the data is presented in tables, and would be far more accessible as figures. Few treatment-to-treatment statistics are presented, and these need to be added. The conclusions are also rather sweeping and not always backed up with the data, nor are they particularly novel nor deterministic. Soil properties, plant communities, and microbial communities all co-vary from plot to plot, so only speculative conclusions can be drawn on the processes driving these patterns.

*Thank you for these suggestions, as noted above and detailed below we have changed many of the tables to figures, with added statistics, to aid the reader in data interpretation. Additionally, we have shifted the conclusions away from more speculative element and to be more in line with the data presented.*

Specific suggestions for improvement:

INTRODUCTION

Line 44: Define PHC here as well and not just in the abstract.

*Thank for noticing this, we have made the change as suggested (line 43, in the revised manuscript).*

Line 51 - 54: A slightly expanded discussion of how phytoremediation is thought to work would be useful. As is, it is unclear if it is thought that the plant metabolites themselves are directly degrading hydrocarbons, or if plant metabolites are stimulating microbes which then degrade hydrocarbons? Do different plants seem better or worse at improving remediation efforts? Do different microbes? Etc

*We appreciate this comment and have added to the introduction to link how plant metabolites might drive microbial transformation of organic pollutants (line 53-57, in the revised manuscript): "This is because soil bacterial communities which are plant-associated are able to use or transform SPMEs, such as phenols, through diverse groups of broadly specific enzymes such as aromatic ring-hydroxylating dioxygenases (Zubrova et al., 2021). These enzymes have also been implicated in the transformation and degradation of organic contaminants, such as PHCs, which are structurally similar to SPMEs(Leigh et al., 2002; Singer et al., 2003; Musilova et al., 2016; Zubrova et al., 2021)"*

I feel that lines 139-146 should be moved to the end of the introduction to make it immediately clear that the expanded microbial data is the focal element for this paper. This could also be made clear with a table highlighting the prior studies from this site, including what was characterized in each study (soil properties, hydrocarbon levels, plant community, and/or microbial community) and how it was characterized in the past vs. now, perhaps incorporated as a panel into Figure 1.

*Thank you for your suggestion, we have moved these lines as suggested and added a statement to the introduction to clarify the data were published and which are novel to this study (Lines 77-94, in the revised manuscript). Additionally, we have moved all findings that are relevant to this study but previously reported to a new section in the results entitled "3.1 Prior Findings on TPH, Soil Properties, and Plant Communities" (lines 207-220, in the revised manuscript). With these additions which make it very clear in two sections which portions of the data set have been previously published, we don't find the addition of another table or figure to highlight the differences between new and previously published data are necessary.*

METHODS

Line 111: Define TPH acronym

*As recommended, we defined the acronym here (line 83, in the revised manuscript).*

Lines 111 – 114: This may fit better moved to the results section.

*Thank you for this suggestion, we have moved these results to the new section "3.1 Prior Findings on TPH, Soil Properties, and Plant Communities" (lines 207-220, in the revised manuscript)*

Table 1: As this data is not new to this study, I think it should be supplemental. I also think it would be better presented as a % reduction from the time 0 TPH ppm value rather than raw values. Stats should also be included so the reader can clearly see where statistical differences between treatments lie.

*Thank you for this comment, we have added an overall percent reduction and statistics to the table. We have moved the table to the supplemental materials (Table S2). While these data have been previously published, they were included as a histogram in Leewis 2013 publication, and therefore the data with standard error add important detail to those previously published. To avoid readers assumption that these data might be new to this publication, the table caption clearly states that the data have been previously published and are reaggregated from the previous publications.*

Section 2.4: It would be useful to add a brief description of how MPN is conducted to give readers the general idea of the method without needing to look up another paper.

*As suggested, we have added a brief description with details of the MPN method (lines 148-158, in the revised manuscript).*

General note: treatment "p1" is sometimes inconsistently labelled instead as "p" (e.g. line 111, Figure S1, etc.)

*Thank you for noticing this inconsistency, we have corrected the labelling throughout the manuscript.*

RESULTS

I suggest that you begin with a section like "Prior findings on TPH, Soil properties, and plant community" combining condensed versions of lines 111-114, section 3.1, and most of section 3.2. This context is important when interpreting the microbial data and ought to be summarized briefly, but it should be clear that it has been published and discussed in greater depth before.

*Thank you for this suggestion, we have added the suggested section as "3.1 Prior Findings on TPH, Soil Properties, and Plant Communities" (lines 207-220, in the revised manuscript), this summarizes the relevant previous findings and clarifies what has been previously published.*

Lines 207 – 210: Did treatment make any difference in the % reduction of TPH?

*Thank you for this suggestion, no one treatment was associated with an overall greater loss in TPH and we have added this to the results section on TPH (Line 227-229, in the revised manuscript): "However, after 15 years of treatment no one treatment resulted in significantly more reduction of TPH concentrations (P > 0.05; Table S2)."*

Table S2: The soil textural data (sand, silt, and clay) that was published in Leewis et al 2013 should be provided in Table S2 as it is referenced later. Was the table cut off? Additionally, stats should be displayed so readers can clearly see significant differences between treatments.

*As the reviewer noted, this data was mistakenly cut off from the page due to formatting in the first submission. We have re-formatted the table and added the relevant statistics to Table S3.*

Lines 213 – 214: Looking at the sand, silt, and clay percentages in Leewis et al 2013, it seems like sand contents are comparable in CO vs. DE soils, and actually clay content was somewhat higher in CO soils. How does this translate to CO being considered coarser in texture? I do see that CEC is lower in CO soils, but while CEC and texture are related CEC is also influenced by many other factors (type of clay, pH, etc). It was mentioned that CO came from a gravel pad – was gravel % included in the textural analysis, or lumped in with sand?

*The reviewer makes an excellent point, and we need to be more precise as to the reasoning that we used the term "coarse soil". Unfortunately, the % gravel was not included in the 2011 textural analyses. We have clarified in the text that the soils came originally from a gravel pad (line 227, in the revised manuscript), and replaced "coarse" with "gravel" or similar throughout.*

Lines 230 – 232: This ground cover % data is novel to this study, to my understanding, and should be kept separate from the "Prior findings" section. Expanded discussion of patterns in Figure 2 would be helpful.

*Thank you for this suggestion, we have moved the ground cover data to clarify that it is new to this paper, (e.g. lines 247-253, 280-285, in the revised manuscript), and expanded the discussion of these patterns (lines 427-430, in the revised manuscript).*

Figure 2: For visualization, it would be useful to add a panel with stacked bar charts for each treatment type showing the breakdown of vegetation types. The current panels should also be kept as they are useful for understanding statistical differences.

*Thank you for this suggestion, we have added this information as supplemental figure 1.*

Line 250: Define PHC acronym

*This acronym was previously defined in the introduction (line 43, in the revised manuscript), therefore we have not added an additional definition.*

Figure S1: I feel this should be a main text figure, as it is the most direct summary of one of the three new pieces of data being presented. Figures S1 and S2 could be combined into a two-panel main text figure, for instance. Stats should be displayed for both as in Figure 2 so the reader can clearly see differences between treatments.

*Thank you for this suggestion, we have added the statistics to this graph and moved it into the main text (Figure 3) for easier access by the reader.*

Line 258: "weak inverse relationship" – this was not statistically significant for either CO or DE soils.

*The reviewer makes a good point, that although the relationship trends in a certain manner, it is not significant. We have changed this sentence to clearly state that it is a non-significant trend and moved the figure to the supplemental information (Figure S2). Lines 275-277: "There was a weak trend between counts of DDM and TPH concentrations in both soil types in which increased counts associated with decreased TPH concentrations, although this relationship was not significant in either crude-oil or diesel-contaminated soils (Fig. S2A & B)."*

Lines 260 to 265: Pearson and p-values should be displayed in the appropriate panel in Figure 3.

*Thank you for this suggestion, we have added Pearson and p-values to the figure panels and moved this figure to the supplemental information (Figure S2).*

Line 262: "number of plants on the site" – what specifically does this mean? The number of individual plants counted? The number of different plant species identified? % plant cover vs. bare ground? Why is a graph representing this not included in Figure 3?

*Thank you for this question, we have clarified this to state that the comparison is between percent coverage of plants on the plots and DDM and added the graphs as supplemental figure 3. Line 277, in the revised manuscript: "there was a positive relationship between DDM and the percent of vegetation coverage in all plots (i.e. no bare ground), although this relationship was only significant in crude oil-contaminated soils (CO: Kendall's $\tau$ = 0.3512, P = 0.0025; DE: Kendall's $\tau$ = -0.0367, P = 0.7623)."*

Line 264: Why did you only investigate the relationship between DDM and tree cover, not grass, forb, or bare cover as well? Were any of those relationships significant?

*Thank you for this comment, we did investigate these other mentioned relationships, but none were significant and we did not include that information in the initial submissions. However, we recognize that it is important to understanding the data set to include this information, we have changed this section to include all of the percent cover data (Lines 284, in the revised manuscript): "There was no statistically significant relationship found between percent coverage of grasses or forbs and DDM in either contamination type (P > 0.06)."*

Table S4: This would be better displayed as boxplots for total biomass and potentially with stacked bar charts for the remaining columns, normalized to percentage of total PLFA for that sample so that differences in overall bacteria:fungi:protozoa ratios can be easily visualized. This could yield more patterns for discussion in the results. Again, stats should be included to aid comparison of treatments.

*Thank you for this suggestion, we have added the total microbial biomass as suggested in a box plot (Figure 3B), and translated the original supplemental table to a graph with statistics (supplemental Figure S4).*

Line 276: In some cases, but for instance diesel c1 vs. diesel p2f is probably not significant. Stats are needed to back up this statement.

*Thank you for noticing this, we have reformulated this section entirely and added updated statistics to be clearer about which treatments were different (line 296-302, in the revised manuscript).*

Line 278: "The same held true for all other individual PLFA biomarkers" – unclear what you mean. There are more fungi, more actinobacteria, etc in fertilized vs. unfertilized? That is unsurprising given the differences in total PFLA amounts. Ratios of relative abundance of these different groups would be more interesting to discuss.

*Thank you for this suggestion, we have completely reworked this section, re-ran the statics to be very clear about which grouped biomarkers are different, and refocused the section on the relationship between total microbial biomass and precent vegetation cover (line 296-310, in the revised manuscript).*

Line 281-282: Not sure the data fully support this statement. E.g. crude oil c1 has high microbial biomass. Clearly displayed stats in Figure S2 would help clarify.

*As stated above, we have rewritten this section (line 296-310, in the revised manuscript), added statistics to this figure, and moved it to the main text (Figure 3).*

Lines 299 – 307 and Figure 4A-B: Because the PFLA community composition data is so coarse already (only 6 different types of microbes, unless for this ordination you separately considered all of the different individual biomarkers listed in Table S1 rather than aggregating them?), I don't think doing an ordination or vector fitting adds anything that wouldn't already be captured in a figure version of Table S4.

*Thank you for this comment, for the ordinations we did not group the biomarkers (as in Table S1) but included them in the multivariate analyses separately. We have added clarification to the methods line 196, in the revised manuscript) and results (line 317, in the revised manuscript) sections. Although the*

*PLFA data is coarse compared to the sequence data, it provides more details regarding the soil community beyond just the prokaryotic community targeted by the primers we used. Therefore, we feel including an ordination of the "total soil community" provides more information than the stacked bar chart, and a more holistic view of the soil community and the impacts of soil type and initial treatment.*

Line 218-319: "significant influence of soil type" – It may be useful to add a panel displaying an NMDS of all samples in this study together, perhaps with crude oil and diesel differently colored, to show separation of communities based on soil type, in addition to panels C and D.

*Thank you for this suggestion, although we had discussed how the communities differ by soil type it was not a figure that was apparent throughout this (or the other) manuscript. We have added a supplemental ordination figure (Figure S5) with both soil types represented to show delineation of the communities from each soil type in both PLFA and 16S sequence data.*

Table S5: This would be better displayed as boxplots with stats between treatments clearly labelled. Could be combined as panels with Figures S1 and S2 into one figure.

*Thank you for this suggestion, we had changed the table into a box plot which more clearly shows the differences between soil type and variability within treatments (supplemental figure 6).*

Figure S3 discussion: is there any functional significance to these families?

*As suggested, we have added this to the discussion (lines 440-444, in the revised manuscript).*

Differential abundance analyses of the 16S data would contribute by identifying specific OTUs associated with particular treatments.

*Thank you for this comment, we have added this analysis to the results (lines 366-373, in the revised manuscript) and discussion (lines 440-446, in the revised manuscript).*

DISCUSSION:

Lines 356 – 357: "influenced the continuing contaminant disappearance" – the stats to back up this statement are not included in Table 1.

*Thank you for pointing this out, we have revised the text to deemphasize this finding, which only was marginally significant (line 386-389, in the revised manuscript)*

Lines 387 – 389: The stats to back up this statement are not included in Table S5.

*These statistics associated with these statements are found in the newly revised Figures S5 and S6, along with Table S2.*

Lines 399 – 400: "the percent coverage of vegetation rather than individual plant species" – Did you incorporate the species level plant data to try to explain the 16S or biomass data? If not, I would leave this comparison out. Percent cover mattered, but without an analysis of individual plant species impact on bacterial diversity we can't say whether this mattered less.

*We have rewritten and clarified this sentence to focus on the percent cover data as suggested (line 428-435, in the revised manuscript*

Line 408 - 409: "number of culturable microorganisms able to degrade TPH was most strongly associated with the plant communities present" – this data was not fully displayed. There was only one positive association with tree cover.

*Thank you for this comment, we have rewritten this sentence to better reflect the data and statistics presented (line 436-440, in the revised manuscript).*

Line 411 - 412: These findings were generally not significant.

*Thank you for this comment, we have rephrased this sentence to fully acknowledge the lack of significance in these relationships (line 439, in the revised manuscript).*

Line 418 – 420: Native vs. non-native plants were not a focus of this study – was this a finding from 2013?

*Thank you for this comment, we have rephrased this sentence to discuss the relationship with coverage trees and indicate that those trees which colonized the site were all native Alaskan trees, as found in the Leewis 2013 study (line 451-453 in the revised manuscript).*

Line 432 – 434: Initial planting of annual vs. perennial plants was not the focus of this study – both plants used were annuals and had also disappeared by the 2011 check in.

*Thank you for this comment, we have removed the reference to annuals or perennials from this sentence (line 466,  in the revised manuscript).*

In general I think the discussion should be re-focused to the microbial data. Do any of the same bacterial taxa from the heatmap or a future differential abundance analysis show up in other phytoremediation studies? Are the microbes identified as important in short term phytoremediation studies still present here? How do microbial diversity levels or biomass levels compare with other phytoremediation studies? For other long-term phytoremediation studies even if they didn't cover all of the different data covered here, are there parallels to be drawn? There are several threads that could be discussed to help contextualize the data.

*We thank the reviewer for these comments and suggestions, as noted above, we have added the suggested analyses (lines 366-373, in the revised manuscript) and discussion points  (lines 440-446, in the revised manuscript) to discuss the microbial community data and draw further comparisons with the literature to help contextualize our data more fully. These suggestions have greatly helped to improve the manuscript.*

---

## Referee Report (RR1)

The authors have addressed all comments in the original review well and this version is much more clear and impactful.

I just have one small additional note – I noticed in Table S3 that some of the stats for nitrate, P, and K indicate that none of the treatments are significantly different from each other (all "a") despite vastly different means (e.g. max 158.67 vs. min 1.00 for Crude oil P). Was this an error?

---

## Author Response (AR2)

*Response to Reviewer Comment 2* | EGUSPHERE-2023-2097
**"Long-term legacy of phytoremediation on plant succession and soil microbial communities in petroleum-contaminated sub-Arctic soils"**
Mary-Cathrine Leewis, Christopher Kasanke, Ondrej Uhlik, Mary Beth Leigh

*We would like to thank the reviewer for their thoughtful review of our manuscript. The original reviewer comments are written in* black *and our reply in* **blue and italics**.

The authors have addressed all comments in the original review well and this version is much more clear and impactful.

I just have one small additional note – I noticed in Table S3 that some of the stats for nitrate, P, and K indicate that none of the treatments are significantly different from each other (all "a") despite vastly different means (e.g. max 158.67 vs. min 1.00 for Crude oil P). Was this an error?

*Thank you for your detailed re-examination of the manuscript. We have verified these statistics, and they are correct as presented in Table S3 -despite the large differences in means, there are not significant differences between treatments for nitrate, P, or K. The data are non-parametric and were analyzed with a Kruskal-Wallace and Mann-Whitney U pair-wise post-hoc test. We agree that because these means are vastly different, one might expect that there would be some significance, perhaps because the test converts data to ranks prior to testing, this method could involve an over correction and therefore the differences are not significant.*

---

## Author Response (AR3)

*Response to Reviewer Comment 1* | EGUSPHERE-2023-2097
**"Long-term legacy of phytoremediation on plant succession and soil microbial communities in petroleum-contaminated sub-Arctic soils"**
Mary-Cathrine Leewis, Christopher Kasanke, Ondrej Uhlik, Mary Beth Leigh

*We would like to thank the reviewer for their thoughtful re-review of our manuscript. The original reviewer comments are written in* black *and our reply in* **blue and italics**.

I recommend accepting the article as the authors have satisfactorily addressed my previous comments and suggestions. There are a few minor details that could still enhance the article, but overall, the revisions meet the required standards. Below, I detail my responses to the authors' modifications:

Script Accessibility: The addition of a GitHub repository for the scripts used in the analyses is commendable. This significantly enhances the study's transparency and reproducibility.

Pyrosequencing Data Clarification: The authors provided a justification for the relevance of pyrosequencing, supported by a reference to a comparative study demonstrating its comparability to newer NGS technologies. The inclusion of specific literature references strengthens the manuscript. However, discussing a broader range of studies could provide stronger support for the continued use of pyrosequencing.

*Thank you for your detailed re-examination of the manuscript. We appreciate the need for stronger support for the use of this outdated technology and have added more information to the methods section to clarify potential issues associated with the use of pyrosequencing and added more examples from the literature of studies that rely on pyrosequencing to advance the understanding of environmental microbial communities (lines 173-177, in the revised manuscript). We would like to be clear that by presenting pyrosequencing data we are not advocating for the continued use of pyrosequencing in the generation of new data, but that data generated by older technologies can continue to hold value for the field of microbial ecology as long as readers and researchers understand the limitations of data generated by these older methods.*

Enhanced Figure Visualization: The effort to include color and shape variations in most figures improves readability and data interpretation. It is recommended that Figures 2 and 3 also receive these updates to maintain consistency across all figures and enhance the overall accessibility of the visual data.

*Thank you for this suggestion, we had added the colour and shapes scheme to figures 2 and 3 to aid in data interpretation and maintain consistent coloration throughout the manuscript.*

Comparison of Concentrations: The updated presentation of Total Petroleum Hydrocarbons (TPH) data and its comparison with findings from other studies enrich the discussion, adding value to the manuscript.

The manuscript has significantly improved with these revisions. The introduction now has a clearer focus, and the discussion section more effectively details the microbiome taxa identified. Additionally, the methods section has been enhanced with a comprehensive description of the processes used to enumerate diesel-degrading microorganisms. These improvements collectively bolster the manuscript's scientific rigor and its contribution to the field.